# Difficulty and pleasure in the comprehension of verb-based metaphor sentences: A behavioral study

**Patrick J. Errington** [1]*, **Melissa Thye** [2], **Daniel Mirman** [2]

1 School of Literatures, Languages and Cultures, College of Arts, Humanities and Social Sciences, University of Edinburgh, Edinburgh, United Kingdom, 2 School of Philosophy, Psychology and Language Sciences, College of Arts, Humanities and Social Sciences, University of Edinburgh, Edinburgh, United Kingdom

☉ These authors contributed equally to this work.
* p.errington@ed.ac.uk

**Data Availability Statement:** All stimulus sets, study design information, and raw data are available from the project Open Science Framework database (https://osf.io/hjcyd/) DOI: (10.17605/OSF.IO/HJCYD).

## Abstract

What is difficult is not usually pleasurable. Yet, for certain unfamiliar figurative language, like that which is common in poetry, while comprehension is often more difficult than for more conventional language, it is in many cases more pleasurable. Concentrating our investigation on verb-based metaphors, we examined whether and to what degree the novel variations (in the form of verb changes and extensions) of conventional verb metaphors were both more difficult to comprehend and yet induced more pleasure. To test this relationship, we developed a set of 62 familiar metaphor stimuli, each with corresponding optimal and excessive verb variation and metaphor extension conditions, and normed these stimuli using both objective measures and participant subjective ratings. We then tested the pleasure-difficulty relationship with an online behavioral study. Based on Rachel Giora and her colleagues' 'optimal innovation hypothesis', we anticipated an inverse U-shaped relationship between ease and pleasure, with an optimal degree of difficulty, introduced by metaphor variations, producing the highest degree of pleasure when compared to familiar or excessive conditions. Results, however, revealed a more complex picture, with only metaphor extension conditions (not verb variation conditions) producing the anticipated pleasure effects. Individual differences in semantic cognition and verbal reasoning assessed using the Semantic Similarities Test, while clearly influential, further complicated the pleasure-difficulty relationship, suggesting an important avenue for further investigation.

## 1. Introduction

'The new dawn blooms as we free it' [1]. That line, like many others in Amanda Gorman's famous poem 'The Hill We Climb', is striking. First performed to great acclaim for US President Joe Biden's inauguration in January 2021, her poem quickly became one of the most celebrated and influential pieces of writing in recent years. But more than the circumstances, the tumultuous politics surrounding its recital, more even than Gorman's riveting performance, the language itself produces a kind of thrill.

**Funding:** This study was funded by a grant from the Research Adaptation Fund from the University of Edinburgh's College of Arts, Humanities and Social Sciences awarded to Patrick J Errington and Daniel Mirman. The funders had no role in study design, data collection and analysis, decision to publish, or preparation of the manuscript.

**Competing interests:** The authors have declared that no competing interests exist.

All the same, the poem's language, like that of many poems, can hardly be said to be easy to understand. (Melanie McDonagh [2], writing in the *Spectator*, opined that the poem is in fact *too* difficult.) A line like 'The new dawn blooms as we free it' is undoubtedly more challenging to comprehend than any (egregiously blunt) literal paraphrase like 'The future will be good'. Why use this 'difficult' language, then? Why can't the poet, as some might complain, 'just say what she means'? After all, things that are easier tend to be more pleasurable. What then is the link, if any, between the particular kind of difficulty encountered in expressive language and the 'affectiveness', the pleasure even, that many experience when reading a piece of writing like Gorman's?

In the present study, we hypothesised that there is such a connection between sentence processing difficulty and pleasure, particularly when it comes to figurative language like metaphor. Specifically, we suggested that pleasure induced by reading metaphorical sentences of increasing degrees of difficulty would also increase, peaking at a certain point before then falling as stimuli sentences become too difficult to resolve. In this, we were informed by the 'optimal innovation hypothesis' put forward by Rachel Giora and her team [3, 4], which suggests that pleasurability is sensitive to what they claim to be an optimal degree of innovation of a given stimulus. A stimulus can, they suggest [4], be considered optimally innovative if it provokes a nondefault response, 'which differs from the default response(s) associated with it, both quantitatively and qualitatively', all the while 'allowing for the automatic recoverability of the default response(s) related to that stimulus, so that both the default and nondefault responses may be weighed against each other, their similarity and differences assessable' (p. 10).

There is a common assumption that it is the figurativeness of poetic language—its use of metaphor, among other non-literal language—that creates difficulty in comprehension. 'The new dawn blooms as we free it' is indeed hardly literal—dawn cannot literally 'bloom' any more than it can be 'freed'. Moreover, 'dawn' here suggests a vision of the future rather than a literal daybreak. Yet such figuration is by no means uncommon. An enormous amount of everyday thought and language is metaphorical [see e.g., 4–12]; we '*run* for office', '*grasp* meanings', '*raise* problems', and none of those strictly literally. Moreover, it is not always the case that metaphor comprehension involves a more lengthy and complicated process than that of literal language [cf. 13, 14]–an attempt at a literal interpretation of the sentence does not always precede a figurative interpretation [see, e.g., 15–17].

Another suggestion is that difficulty is the result of unfamiliar or 'novel' language. Again, this is true of Gorman's poem and enduring trends in literary and artistic theory—beginning, perhaps, with the Russian formalists but with a lineage tracing back as far as Aristotle—also contend that the defining feature of art is novelty and that this novelty slows perception and comprehension by inducing difficulty [see, e.g., 18]. According to formalist Viktor Shklovsky [18], by 'defamiliarizing' or 'making strange' (*ostranenie*, in Russian) habitual experiences and perceptions, art makes it so that 'one may recover the sensation of life; it exists to make one feel things, to make the stone *stony*' (p. 4).

However, any increased 'affectiveness' or 'pleasure' cannot be the result of novelty alone. For one, one encounters unfamiliar sentences every day and yet has little difficulty comprehending them—indeed, many of the sentences in this very article are likely novel to many readers. It must be recalled that, in the 'defamiliarization' that Shklovsky [18] and other formalists advocate, the familiar must nevertheless remain perceptible despite whatever manipulations the artist has subjected it to, as some theorists have emphasised more recently [see, e.g., 19]. In other words, in making 'the stone *stony*', one cannot change it so much that it is no longer recognisable as a stone.

Much of this aligns with Rachel Giora and her team's 'optimal innovation hypothesis' [3, 4]. The most recent iteration of this hypothesis [4] proposes that a pleasurable experience results

from the altering of a stimulus enough that it elicits a novel, nondefault response, yet all the while continuing to elicit the default response: 'The result is that both interpretations [default and nondefault] are entertained and interact' (p. 10). This, they suggest, is at the heart of the pleasure felt in experiencing art, hearing jokes, and reading poetry—they are pleasurable not despite the fact that understanding such experiences is more difficult, but because of it.

The optimal innovation hypothesis adds to a long tradition of research regarding the relationship between stimulus complexity and aesthetic appreciation [see 20 for a recent review]. While research in this tradition has tended toward visual or auditory stimuli rather than linguistic, Giora et al.'s hypothesis can be considered an extension of Berlyne's 1971 proposed inverted U-shaped relationship between stimulus complexity (producing processing difficulty) and aesthetic experience (pleasure), with the highest preference for stimuli of an intermediate level of complexity [21]. Attempts to test Berlyne's hypothesis have yielded complicated findings, with evidence both supporting [see e.g., an overview of U-shaped preferences in music, 22] and contradicting Berlyne's prediction [for evidence of linear relationships, see 23, and non-inverted U-shaped relationships, see 24]. These conflicting empirical results have been suggested to indicate not only crucial differences in how complexity is defined, measured, and manipulated, but also the ways that individual differences might shift this relationship [20].

For their part, Giora et al. [3] specify that optimal innovation is more than any simple 'variant' or 'complication' of a given stimulus. Both a familiar (e.g., 'A piece of paper') and a variant (e.g., 'A single piece of paper') 'refer to the same concept [. . .] to which the variant stimulus contributes no qualitatively different response' (p. 117). As such, however, it becomes unclear if a novelised metaphor, like Gorman's [1] 'The new dawn blooms', could be considered optimally innovative or simply a variant: both the line and a more familiar alternative (e.g., 'The new dawn breaks') would provoke the same default interpretation (e.g., 'The future is here'). Gorman's is a significant change from the familiar phrase, in that 'bloom' is generally more associatively positive than 'break', but is it sufficiently different to be more than just a variant? 'A single piece of paper' is also quite different from 'A piece of paper', underscoring its singularity, whilst still provoking a default interpretation. However, the majority of examples of optimal innovations Giora et al. [3] provide are puns—e.g., 'a *peace* of paper' as an optimal innovation of 'a piece of paper' (p. 117), and as such invoke an entirely new conceptual domain from the default ('a piece of paper'). Does a metaphor variation, like exchanging 'bloom' for 'break', have the similar effect of invoking a whole new conceptual domain?

Novel variations of familiar metaphors may be ideal candidates for optimal innovation, despite the fact that Giora et al. [4] explicitly separate defaultness from figurativeness. As suggested in Lakoff and Johnson's *Metaphors We Live By* (1980) [9] and quantified by Pollio et al. [11], a considerable amount of everyday language can be considered metaphorical, with many of these metaphors taking highly conventional forms and thereby eliciting an especially strong default response. While Lakoff and Johnson claim that all conceptual metaphors, regardless of conventionality, involve an active mapping from one conceptual domain onto another (i.e., that comprehending 'I run for office' involves a mapping of the domain of 'running' onto the domain of 'seeking elected office') [9–10], evidence from recent studies does not bear this out [25, 26]. 'Highly conventional metaphors do not appear to require online access to conceptual mappings', Holyoak and Stamenković [27] summarise; '(i.e., such mappings are even easier than "automatic"—they are not performed at all)' (p. 655). Therefore, highly conventional metaphors like 'I run for office', would *only* recruit the default interpretation (e.g., 'I am seeking elected office') and require no mapping of two conceptual domains.

Still, some aspect of that 'mapping' must remain because novel variations of conventional metaphors like 'I run for office' are more permissible than variations of fully lexicalised metaphors like 'I kicked the bucket'; 'I dash for office' is still comprehensible as 'I am seeking

elected office' in a way that 'I punt the bucket' is likely to be interpreted only literally. Notably, variation like 'I eat for office' does not provoke the default interpretation at all, since 'eat' is a completely different conceptual domain than 'run'. A variation like 'I dash for office', however, is novel whilst remaining within the same conceptual domain as 'run'. Such a variation should theoretically invite both a default interpretation ('I am seeking to be elected') while re-activating the latent conceptual mapping. Thus, it should provoke a nondefault meaning in the form of the source/vehicle of the metaphor ('I am [literally] dashing / to get elected'). The same should be true of extensions to those metaphors based on the critical verb: e.g., 'I run for office but get tripped up along the way' is resolvable via the familiar metaphor in a way that 'I run for office but get so full I can no longer move' (whose extension is outside the domain of 'running') is not. As a result, such optimally innovative metaphors should produce a pleasurable response akin to the puns Giora et al. [3, 4] have tested.

Nevertheless, the balance that allows both default and nondefault interpretations is likely to be quite delicate, and even small changes could upset that balance. We hypothesised that making the source/vehicle (in the form of the verb or extension) too domain-specific might over-privilege the nondefault (literal) interpretation and make simultaneously resolving the figurative interpretation too difficult to be pleasurable. In cases like 'I ski for office', it fits the domain of 'competitive physical movement', like 'run' or 'sprint', and can possibly be resolved coherently—unlike 'I eat for office'—but we hypothesised that this would be too much for most readers.

What is more, the mapping and resolution processes involved in comprehending metaphors are, like other cognitive processes, subject to differences between individuals. A metaphor will be differentially difficult for different readers and, by hypothesis, so will the pleasure derived from it, even if the underlying relationship between difficulty and pleasure is the same. Experimental design provides some control of difficulty as a property of the stimuli, but it cannot be ignored that what some people find difficult others will find very easy. Indeed, Reber and his colleagues, have suggested [e.g., 28] that, for visual stimuli at least, an individual perceiver's processing dynamics are the key determinant of aesthetic experience: 'The more fluently the perceiver can process an object, the more positive is his or her aesthetic response' (p. 366). While this would seem to predict a linear relationship between difficulty and pleasure, in the case of our study, the point at which a stimulus metaphor is complex enough to provoke both default and nondefault responses simultaneously (the apex of the inverted U-shape) is likely to depend on an individual's processing aptitude. To that end, in addition to participant assessments of processing difficulty, our study also used a Semantic Similarities Test (SST). This test assesses an individual's ability to identify conceptual mappings between words (a form of crystalized verbal intelligence), an ability that has been shown to influence one's capacity to process both novel and familiar metaphor comprehension [29]. High scores on the SST should correlate with higher ease ratings for all classes of stimuli—familiar, moderately innovative, and very innovative—and should therefore shift the apex of any inverted U-shaped pleasure curve toward the more innovative.

One limitation of current metaphor research is its tendency to focus on 'nominal metaphors' (metaphors expressed in the traditional 'X is Y' formulation), as noted in Holyoak and Stamenković's review [27]. For this reason, and because of what we perceive to be a relative lack of straightforward nominal metaphors in poetry—Gorman's poem being a prime example—and other expressive writing, we have elected to focus on other forms of metaphor. Because conceptual metaphors, like those described by Lakoff and Johnson [9, 10], tend to be expressed in verb-form rather than the nominal, we have chosen to focus on verb metaphors such as 'I run for office' (where the verb 'run' is being used metaphorically, with the nominal metaphor 'elected offices are the finish-line of a race' implied by that verb).

Several studies have suggested that aptness rather than conventionality is a better predictor of metaphor performance [30–32]. In our case, we have elected to use highly conventional metaphors as the base and vary these by either changing the metaphorically employed verb or by extending the metaphor (with extension being related to the operative verb). Since all our stimuli are based on highly conventional metaphors, aptness should be to some degree controlled, even as in progressively more novel variation, that aptness may be less readily apparent. (It should also be noted that aptness is highly context dependent, something that we did not manipulate in this study).

Other research, such as a series of studies by Al-Azary and Buchanan [33] suggest that metaphor comprehensibility can be related to what has been called 'semantic neighborhood density' (SND)–the number and proximity of semantically similar words. The semantic field around a term like 'ski', for instance is much less dense than around a term like 'run'. Finding, in an offline comprehension task, that tenors and vehicles with low-SND were generally more comprehensible than those with high-SND tenor and vehicles, they suggested that the many semantic neighbours of high-SND terms interfere with the computation of a new meaning for that term. However, the metaphors they tested were all relatively novel, nominal (x is y) metaphors, quite unlike the verb-based, familiar (and varied) metaphors we examined here. Moreover, while they also examined the interaction of SND with tenor concreteness, in our case, the tenors in our study are all abstract. While SND was not calculated or manipulated in our study, its potential to play a role in comprehensibility of novel metaphors is worth bearing in mind for future studies.

As it stands, present study examined the optimal innovation hypothesis [3, 4] using novel variations of familiar metaphors. This complements prior studies that focused on comprehension of completely novel metaphors [e.g., 33, 34], though we believe variations of familiar metaphors are actually more common in both everyday communication and highly specialised communication like poetry. To make this possible, we developed a broad set of stimuli phrases, matched for psycholinguistic characteristics, and assessed on familiarity, ease of interpretation, figurativeness, and imageability. These stimuli and their characteristics are available at https://osf.io/hjcyd/ as a resource for future studies of non-nominal metaphor comprehension. The present study examines the relationship between comprehension difficulty and pleasure using a combination of experimentally manipulated stimuli and observational measures of individual differences between participants. Individual differences were measured using the Semantic Similarities Test (SST), to begin to unpick how the aptitudes and characteristics of individuals might influence the difficulty-pleasure relationship.

## 2. Stimulus set development and norming

In order to examine the relationship between pleasure and difficulty in the comprehension of verb-based metaphors (Experiment 1), we first developed and normed a set of sentence stimuli using objective measures and subjective ratings. All human data collection was conducted online using Qualtrics.

### 2.1. Stimulus development

A set of 372 English-language stimulus sentences were developed, in part from conceptual metaphor types compiled in Lakoff, Espenson and Schwartz [35]. These were organised into 62 sets of 6 variation categories. The starting point was a familiar conceptual metaphor and a minimally different literal sentence. Novel metaphoric sentences were derived from the familiar metaphor in two ways: by changing the critical verb or by extending it with an additional phrase. These derivations were also done to two degrees: an 'optimal' innovation that was

**Table 1. Two example sets of sentences.**

| Literal Sentence | Familiar Metaphor | Optimal Verb | Optimal Extension | Excessive Verb | Excessive Extension |
|---|---|---|---|---|---|
| I grasp the railing | I grasp the meaning | I brush the meaning | I grasp the meaning and shake it vigorously | I tickle the meaning | I grasp the meaning and swing on it |
| I gather my sticks | I gather my strength | I amass my strength | I gather my strength until I can't hold it any more | I pile my strength | I gather my strength into a bundle and then tie it |

moderately close to the familiar metaphor and an 'excessive' innovation that was substantially farther from the familiar metaphor. Table 1 shows two example sets of 6 sentences.

The first-person subject ('I') was applied across all sentences, to avoid gender and animate/inanimate distinctions of the English third-person subject ('he'/'she'/'they'/'it'), and the verb tense was uniformly present tense. Six sentence sets were in passive voice (e.g. 'I am transported by a poem'); 2 sentence sets employed prepositional variations rather than verb (e.g. 'I am in trouble' / 'I am nearing trouble'); 2 sentence sets employed adjectival present participle variations (e.g., 'I have a burning desire' / 'I have a smouldering desire'); and 1 sentence set had a subject other than 'I' but maintained the first-person voice ('My hand/plan hits a brick wall'). These variations were normed for potential use, but not included in the stimulus set used in Experiment 1.

## 2.2. Objective measures

The number of words in each sentence was matched within sets, with literal, familiar metaphoric, optimal verb, and excessive verb sentences all containing the same number of words (ranging from 4 to 6 words); both optimal and excessive metaphor extensions had an equivalent number of words within each set (ranging from 9 to 14). In order to derive an objective estimate of semantic distance between the various critical verbs across the different conditions (familiar, optimal, excessive) and between those critical verbs and the abstract nouns they metaphorically modify within each condition, the cosine similarity between (1) the critical verbs across the condition variations (i.e., *grasp—brush)* and (2) the critical verb-noun words within condition (i.e., *grasp–meaning*) was calculated using pre-trained vector-based representations of word meaning (word2vec). Phrase frequencies were generated for literal, familiar, optimal verb and excessive verb variations using the Google NGram search engine implemented with the ngramr package in R [36]. Phrases were modified to include wildcard tags for variations in determiner use and inflection to include subtle phrase modifications in the frequency calculation. As anticipated, familiar metaphor and literal sentences were more frequent than the optimal or excessive verb variations (i.e., the novelised metaphors were indeed novel; see bottom of Fig 1).

## 2.3. Subjective ratings

A total of 94 adult participants were recruited: 74 via the University of Edinburgh's SONA student participant recruitment program and 20 via Prolific. Participants either received course credit or £3.50 upon completion of the 30-minute study. To prevent sentence variations from the same set appearing in consecutive trials, the sentences within the stimulus sets were assigned to one of six lists. Each list contained 62 sentences and an equal number of sentences from each variation category. An attention check question was added to each list to assess participant engagement and data quality. These questions instructed participants to select a specific number on the scale (i.e., 'Select number six for this question'). The presentation of the

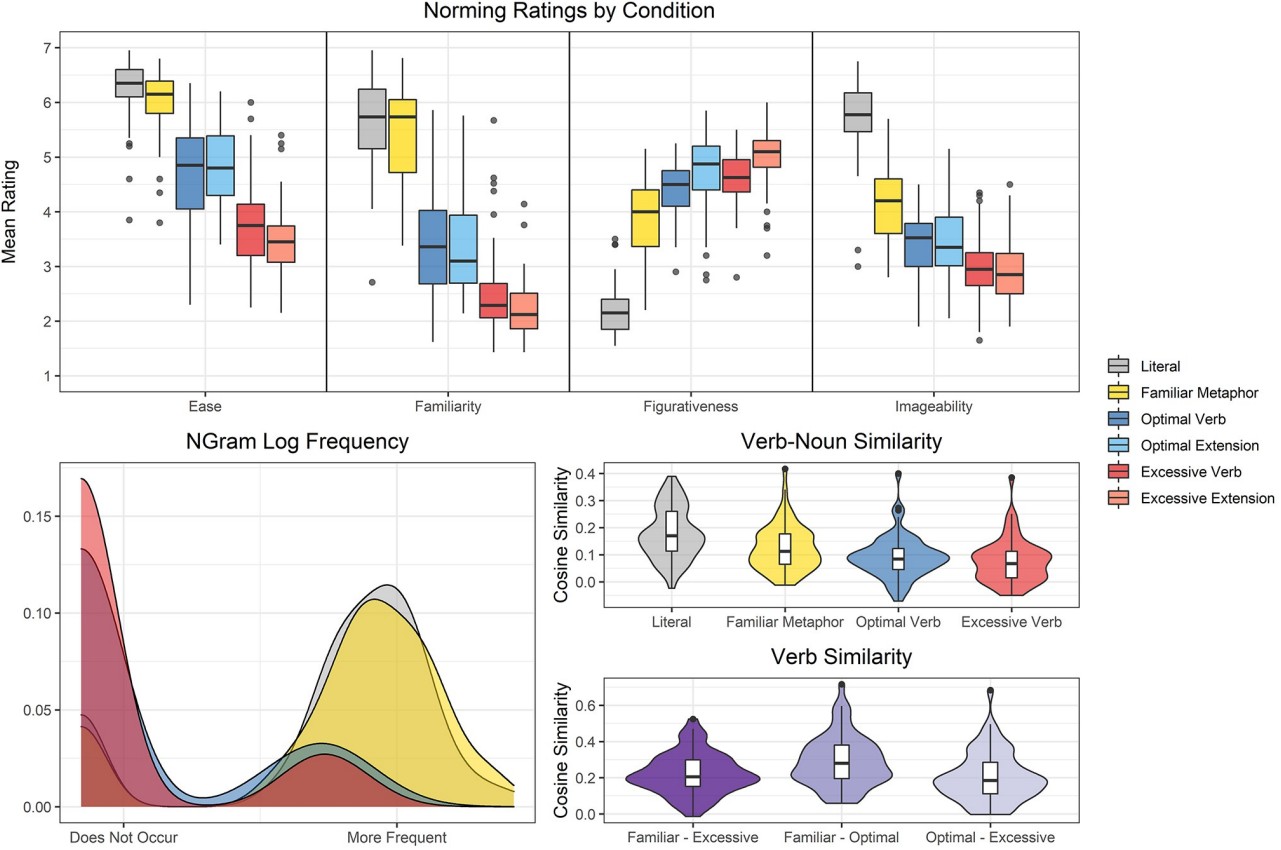

**Fig 1. Objective and subjective measures for the 6 sentence categories.** The subjective ratings of Ease (n = 20), Familiarity (n = 21), Figurativeness (n = 20), and Imageability (n = 20) are shown for each variation category in the top row. The NGram phrase frequency (log-scaled) values are shown in the bottom left panel: the peaks on the left edge indicate 0 frequency (i.e., not found in the corpus) for many of the metaphor variations (particularly the verb variations), the peaks in middle-right indicate moderately high frequencies for the literal sentences and familiar metaphors. Cosine similarity within variation condition (Verb-Noun Similarity) and between variation condition (Verb Similarity) are shown in the bottom right panel.

lists and the sentences within each list were randomized. At the start of the experiment, participants read the description of the property they would rate sentences on and were given at least 3 example sentences with variable ratings illustrating how to use the 7-point scale (full instructions with examples are available on the project OSF page). Participants rated the 372 stimulus sentences on *one* of the four following properties (approximately 20 participants per property):

- **Ease**: How easy the sentence was to interpret on a scale from 1 (very difficult) to 7 (very easy)

- **Imageability**: How quickly and easily each sentence aroused a sensory experience (i.e., a mental picture, sound, texture, or action) on a scale from 1 (no image) to 7 (clear, immediate image)

- **Familiarity**: How familiar the meaning of each sentence seemed or how commonly one might encounter such a meaning on a scale from 1 (very unfamiliar) to 7 (very familiar)

- **Figurativeness**: Whether the event described in each sentence is literal (could actually happen) or whether the sentence likely conveys a more figurative meaning on a scale from 1 (very literal) to 7 (very figurative).

## 2.4 Norming results and discussion

A total of 13 participants were excluded from analysis due to failing more than 3 of the 6 attention checks, resulting in a final sample size of 81. The full stimulus set and the corresponding sentence-level objective and subjective measures are available on the project OSF page [https://osf.io/hjcyd/].

The distributions of the objective measures and the subjective ratings are shown in Fig 1. There is substantial variation within each sentence condition, but the patterns align with the intent of the conditions. Semantic relatedness (cosine similarities) between verbs and nouns was highest in the literal sentences ($Mdn$ = 0.17), followed by the familiar metaphors ($Mdn$ = 0.11), and lowest for the optimal verb ($Mdn$ = 0.08) and excessive verb ($Mdn$ = 0.07) conditions. Cosine similarities between the various verbs in each stimulus set were generally higher than the critical verb-noun similarities ($Mdn$ = 0.19–0.28), suggesting that any difficulty in processing should not be the result of unusual verb choices in any particular condition. Excessive verbs tended to be less similar to familiar metaphor and to optimal verbs than these verbs were to each other. The observed pattern of less within (verb-noun) and between (verb-verb) condition similarity for excessive sentences is consistent with the claim that they are 'excessively' far from the familiar metaphors. The individual phrase frequencies generated by using the Google Ngram data were very low, with phrase frequencies of 0 for 9 familiar metaphors, 10 literal sentences, 47 optimal verb metaphors, and 52 excessive verb metaphors out of the total of 62 for each. This aligned with initial expectations that the familiar metaphors would be the most familiar of the individual sentences, followed by literal sentences, with both optimal and excessive verb metaphors being entirely or very nearly novel.

Participant ratings aligned with these objective metrics. Ease of interpretation, familiarity, and imageability were highest for the literal ($Mdn_{Ease}$ = 6.35; $Mdn_{Familiarity}$ = 5.74; $Mdn_{Image}$ = 5.78) and familiar metaphor ($Mdn_{Ease}$ = 6.15; $Mdn_{Familiarity}$ = 5.74; $Mdn_{Image}$ = 4.20) sentences, followed by the optimal verb ($Mdn_{Ease}$ = 4.85; $Mdn_{Familiarity}$ = 3.36; $Mdn_{Image}$ = 3.53) and extension ($Mdn_{Ease}$ = 4.80; $Mdn_{Familiarity}$ = 3.10; $Mdn_{Image}$ = 3.35) and the excessive verb ($Mdn_{Ease}$ = 3.75; $Mdn_{Familiarity}$ = 2.29; $Mdn_{Image}$ = 2.95) and extension ($Mdn_{Ease}$ = 3.45; $Mdn_{Familiarity}$ = 2.12; $Mdn_{Image}$ = 2.85) variations. Importantly, this was shown to be similar in the case of both optimal and excessive extensions, which we could not determine with objective measures. An inverse pattern was observed for figurativeness ratings which were lowest for the literal sentences ($Mdn_{Figurative}$ = 2.15) and increased for familiar metaphors ($Mdn_{Figurative}$ = 4.00) and the optimal verb ($Mdn_{Figurative}$ = 4.50) and excessive verb ($Mdn_{Figurative}$ = 4.63) variations. Optimal extensions ($Mdn_{Figurative}$ = 4.88) and excessive extensions ($Mdn_{Figurative}$ = 5.10) were rated as the most figurative.

These ratings were used to select a subset of 45 sentence sets for Experiment 1. The 17 stimulus sets that were not included were metaphors that were rated as not very metaphoric (low average figurativeness ratings and/or high imageability ratings; n = 3), optimal verb or extensions that were rated not very innovative (high familiarity ratings; n = 9), excessive verb and extensions that were too easy to understand (high ease of interpretation ratings; n = 3), and the use of adjectival present participle sentences (e.g. 'I have a burning/smouldering desire') as these stood out because they did not involve a subject acting or being acted upon (n = 2). We also opted not to test the literal sentences to simplify the experiment, since these were the stimuli that were being directly manipulated. After all, familiar metaphors were rated as familiar and as easy to comprehend, roughly, as the literal phrases, suggesting that for these phrases the metaphoric interpretation ('I grasp the situation' = 'I comprehend the situation') is equally the default as the literal interpretation of a literal phrase ('I grasp the railing' = 'I hold on with my hand to the railing'). Where, for Giora et al. [3], the default interpretation of 'A piece of paper'

was literal before being altered to produce the simultaneous metaphoric/literal pun, 'A *peace* of paper', for our study the default interpretation is metaphoric with the optimal variation intended to elicit both metaphoric and, to some degree, literal meaning.

## 3. Experiment 1

This experiment was designed to examine the relationship between pleasure and the ease of comprehension of verb-based metaphors using a combination of experimental manipulation (sentence category), subjective ratings (ease, pleasure), and individual differences (Semantic Similarities Test performance). Based on Giora et al.'s 'optimal innovation hypothesis' [3, 4], we anticipated that, as difficulty increased from familiar to excessive variations (both verb variations and extensions), pleasure would form an inverse U-shape, with optimal variations receiving higher pleasure ratings than either the familiar or excessive variations.

### 3.1. Method

Participants were 63 adults, recruited online via Prolific, all with English as their first language, no history of language-related disorders and no history of mild cognitive impairment or dementia. Both the stimulus norming and Experiment 1 were carried out in accordance with an ethics protocol approved by the University of Edinburgh PPLS Research Ethics panel (Ref No. 277-2021/3). As we tested a new set of stimuli and there was not a strong basis for specifying an a priori effect size, a power calculation was not possible. The sample size was determined based on prior studies of metaphor comprehension, which typically tested 30–60 participants [see e.g., 4, 31, 32], and practical limitations on data collection. We hope that providing via OSF the complete set of analysis procedures and code, along with the full stimulus set will allow researchers to determine appropriate sample sizes for future studies.

Participants received £7.25 upon completion of the approximately 1-hour study. Participants were randomly assigned to 1 of 3 experimental groups. The groups differed in the set of sentences shown to participants, but each group of stimuli contained the same overall number of items (n = 75) as well as an equal number of items from each variation category. A couple of sentence variations within a stimulus set may be included in each group, but none of the groups included all variations. Participants rated the sentences within their group on each of the subjective ratings (below), ensuring that sentence-level ratings were within-subject. Similar to the preliminary norming study, sentence variations were assigned to one of five lists. An attention check question with the same structure as those used in the norming study was added to each list. For a given rating (i.e., Ease), the set order and the presentation of the sentences within each list were randomized, but each rating block began with a description of the property and at least 3 example sentences. The presentation order of the subjective rating blocks was counterbalanced such that participants were randomly assigned to one of four presentation orders. (A complete survey file is available on the project OSF page).

Participants in all groups rated the sentences on the following properties (full instructions and examples are available on the project OSF page):

- **Ease**: How easy the sentence was to interpret on a scale from 1 (very difficult) to 7 (very easy)

- **Imageability**: How quickly and easily each sentence aroused a sensory experience (i.e., a mental picture, sound, texture, or action) on a scale from 1 (no image) to 7 (clear, immediate image)

- **Emotion**: How strongly each sentence evoked an emotional response on a scale from 1 (no emotion) to 7 (strong emotion)

- **Pleasurability**: How much the participant liked the way the message was expressed focusing on how effective, satisfying, or powerful the sentence was on a scale from 1 (not pleasurable) to 7 (very pleasurable)

Between the subjective rating blocks, participants completed blocks of individual differences measures, which were presented in a fixed order. These measures included the Semantic Similarities Test (SST), which was manually scored according to the criteria outlined by Stamenković, Ichien, and Holyoak [23].

In developing the test questions, we took particular note of the potential complications raised by Giora et al. [3] resulting from the use of the word 'pleasure' in the assessment of aesthetic experience of reading literary language; because the word is positively associated, we were concerned that it might contribute to a negative bias toward sentences with negative valences or meanings (e.g., 'I am bruised by grief'). While we also note Schindler and her colleagues' recent review of methodologies for measuring aesthetic experience [37], in the case of this study, we, following Giora and her team, were primarily interested in the potential increase in a very broad idea of 'pleasure', aligning more with 'affectiveness' or 'felt experience' [38] provoked primarily by the formal features of the phrase rather than any particular emotion: i.e. if the sentence content was broadly 'sad', did the formal variation make it 'sadder'; if 'pleasant', did the variation make it more pleasant; and so on. To that end, we presented the instructions as follows:

> You will read a series of sentences that have figurative (non-literal) meanings as well as literal meanings. For each sentence, rate on a 7-point scale how much you liked the way the message was expressed. This does *not* mean that you liked the message; rather, you should rate how effective, satisfying, or powerful the sentence was. It might help to think about whether you would enjoy reading such a sentence in a book or poem.

### 3.2. Results and discussion

Data from 3 participants were excluded from analysis due to failing any of the 5 attention checks (n = 1), providing the same response to all items within a block (n = 1), or failing to complete the entire survey (n = 1), resulting in a final sample size of 60 (20 per group; female = 38, male = 22, mean age = 31.32). All participants completed the Semantic Similarity Test and received credit for at least one response ($M$ = 22.12, $SD$ = 6.88).

Data were analyzed using linear mixed effects models implemented with the lme4 package (version 1.1.23) [39] in R (version 4.0.2) [40]. Model parameter $p$-values were obtained using the Satterthwaite method for estimating degrees of freedom via the lmerTest package (version 3.1.2) [41]. Continuous predictors were centered prior to analysis.

A first set of analyses directly compared the metaphor conditions (as fixed effects) with random by-participant intercepts and slopes of variation category and random intercepts of item. The results of these analyses (Table 2 and Fig 2) revealed that both verb and extension manipulations produced monotonic differences in ease of comprehension: Familiar metaphors easier than optimal variations, which were easier than excessive variations. This provides a validation of the manipulation—the "excessive" metaphor variations were indeed more excessive (difficult to comprehend) than the "optimal" metaphor variations. For pleasure ratings, the verb variations elicited an analogous monotonic effect: Familiar metaphors were rated as more pleasurable than optimal verb variations, which were more pleasurable than excessive verb variations. Pleasure ratings for metaphor extensions exhibited the predicted U-shape for extensions: Familiar metaphors were rated as *less* pleasurable than optimal extensions and marginally more pleasurable than excessive extensions.

**Table 2. Parameter estimates (standard error in parentheses) for variation conditions relative to the familiar metaphor condition.**

| Condition | Ease | Pleasure |
|---|---|---|
| Optimal Verb | -1.32 (0.12) *** | -0.27 (0.13) * |
| Excessive Verb | -2.14 (0.16) *** | -0.75 (0.15) *** |
| Optimal Extension | -0.89 (0.12) *** | 0.40 (0.18) * |
| Excessive Extension | -2.03 (0.21) *** | -0.37 (0.20) |

Note: $p < 0.1$,

* $p < 0.05$,

** $p < 0.01$,

*** $p < 0.001$

Although the metaphor variations conditions were intended to manipulate comprehension difficulty, there was substantial variation in ease of comprehending metaphors within each condition. Further, differences between participants in semantic knowledge were also expected to influence comprehension difficulty. Therefore, a second set of analyses assessed how pleasure was predicted by sentence variation category (fixed effect with familiar metaphor as the reference level) and ease of interpretation (Ease fixed effect in Model 1) or individual differences in semantic knowledge (SST fixed effect in Model 2). Model 3 assessed the impact of sentence variation type and semantic knowledge on ease of interpretation. All models included random by-participant intercepts and slopes of variation category and random intercepts of item.

Model 1 results are shown in the top section of Table 3 and the left panel of Fig 3. Pleasure ratings were highest for the optimal extension sentences (they were approximately equal for the other 4 categories of sentences) and tended to increase with ease of interpretation. There was also a significant interaction between ease and sentence category: the positive association between ease and pleasure was weaker for the familiar metaphors than for the other 4 sentence categories (though not statistically significantly different from the optimal extension category).

Model 2 results are shown in the middle section of Table 3 and the middle panel of Fig 3. Participants with higher SST scores (better semantic knowledge) tended to give lower pleasure ratings. However, this was qualified by an interaction: pleasure ratings for the two extension sentence types (optimal extension and excessive extension) were essentially constant across the range of SST performance.

Model 3 results are shown in the bottom section of Table 3 and the right panel of Fig 3. Not surprisingly, ease of interpretation was positively associated with SST scores: participants with better semantic knowledge found the metaphoric sentences easier to understand. Also not surprising (and replicating the preliminary norming results) was that familiar metaphors were rated the easiest to understand, followed by optimal verb and extension sentences, and excessive verb and extension sentences were rated the most difficult to understand. There was also an interaction: SST was most strongly associated with ease of interpreting the familiar metaphor sentences and least associated for the excessive verb sentences (the other sentence categories were intermediate). That is, semantic knowledge appeared to be particularly important for understanding familiar metaphors, but not for making sense of novel sentences.

## 4. General discussion

Our working hypothesis was that as metaphor variants moved farther from their familiar metaphor base, they would become more difficult to comprehend and that pleasure would peak at an intermediate point—where innovation was 'optimal' [3, 4]. The results of both the norming

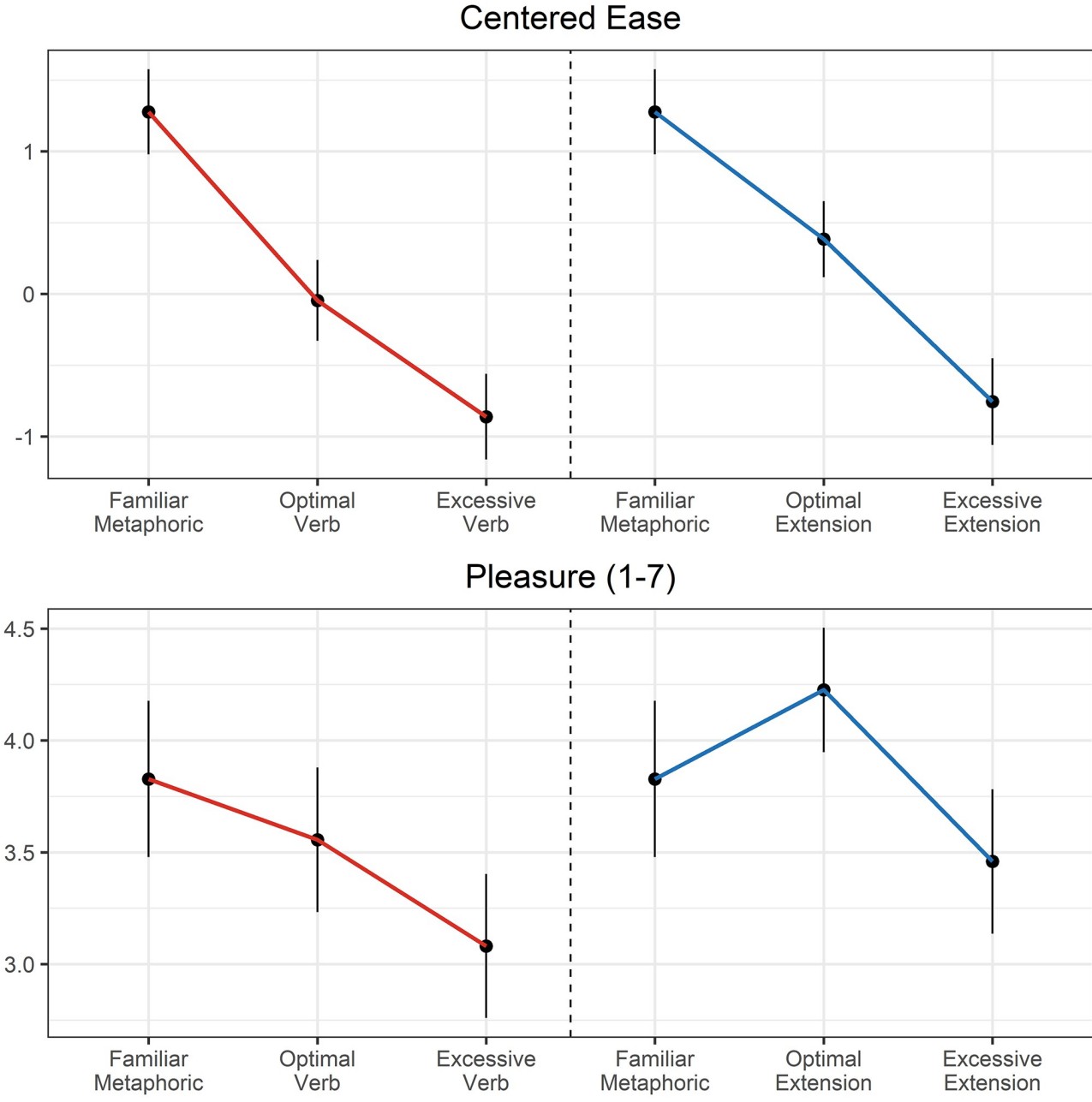

**Fig 2. Ease and pleasure ratings by condition.**

study and the experiment confirmed that our stimulus manipulation elicited the intended ease of comprehension effect: familiar metaphors were rated the easiest to understand, the 'optimal' verb and extension variants were somewhat more difficult, and the 'excessive' verb and extension variants were the most difficult. However, the consequent effect on pleasure ratings were only partly consistent with the 'optimal innovation hypothesis'.

The strongest support came from the variations made by extending the familiar metaphors. The optimal metaphor extensions were rated as intermediate in terms of ease of comprehension (more difficult than the familiar metaphors and easier than the excessive metaphor

**Table 3. Experiment 1 continuous analyses of effect of ease on pleasure ratings (Model 1), effect of individual differences (SST) on pleasure ratings (Model 2), and effect of individual differences (SST) on ease ratings (Model 3).**

| Model 1 | Term | Estimate (SE) | p-value |
|---|---|---|---|
| | Optimal Verb | -0.02 (0.13) | 0.873 |
| | Optimal Extension | 0.54 (0.18) | 0.005** |
| | Excessive Verb | -0.23 (0.15) | 0.125 |
| | Excessive Extension | 0.09 (0.21) | 0.683 |
| | Ease | 0.19 (0.03) | 0.000*** |
| | Optimal Verb x Ease | 0.10 (0.04) | 0.029* |
| | Optimal Extension x Ease | 0.08 (0.05) | 0.103 |
| | Excessive Verb x Ease | 0.13 (0.04) | 0.003** |
| | Excessive Extension x Ease | 0.10 (0.05) | 0.032* |
| Model 2 | | | |
| | Optimal Verb | -0.27 (0.13) | 0.040* |
| | Optimal Extension | 0.40 (0.17) | 0.025* |
| | Excessive Verb | -0.747 (0.15) | 0.000*** |
| | Excessive Extension | -0.37 (0.19) | 0.063 |
| | SST Score | -0.05 (0.02) | 0.034* |
| | Optimal Verb x SST Score | 0.00 (0.12) | 0.822 |
| | Optimal Extension x SST Score | 0.06 (0.03) | 0.020* |
| | Excessive Verb x SST Score | 0.00 (0.02) | 0.855 |
| | Excessive Extension x SST Score | 0.06 (0.03) | 0.030* |
| Model 3 | | | |
| | Optimal Verb | -1.32 (0.12) | 0.000*** |
| | Optimal Extension | -0.89 (0.12) | 0.000*** |
| | Excessive Verb | -2.14 (0.15) | 0.000*** |
| | Excessive Extension | -2.03 (0.19) | 0.000*** |
| | SST Score | 0.05 (0.02) | 0.012* |
| | Optimal Verb x SST Score | -0.03 (0.02) | 0.145 |
| | Optimal Extension x SST Score | -0.02 (0.02) | 0.354 |
| | Excessive Verb x SST Score | -0.06 (0.02) | 0.016* |
| | Excessive Extension x SST Score | -0.04 (0.03) | 0.144 |

*Note.* SE, standard error. Sentence variation conditions are referenced to the familiar metaphor condition.

*$p < .05$.

**$p < .01$.

***$p < .001$.

extensions), but highest in terms of pleasure (higher than familiar metaphors and excessive metaphor extensions, which were rated approximately equally pleasurable). This is consistent with the optimal innovation hypothesis and counter to the typical pattern that pleasure is monotonically associated with ease.

Verb variations followed the more typical pattern of a monotonic relationship between ease and pleasure: familiar metaphors were rated both easier and more pleasurable than 'optimal' verb variants, which were rated both easier and more pleasurable than 'excessive' verb variants. For the 'excessive' sentence types, extensions were rated as being more pleasurable without being easier than verb variants, which is partially consistent with the optimal innovation hypothesis. The broader pattern that metaphor extensions were rated as being more pleasurable than verb variants (among both 'optimal' and 'excessive' sentence types) may be more

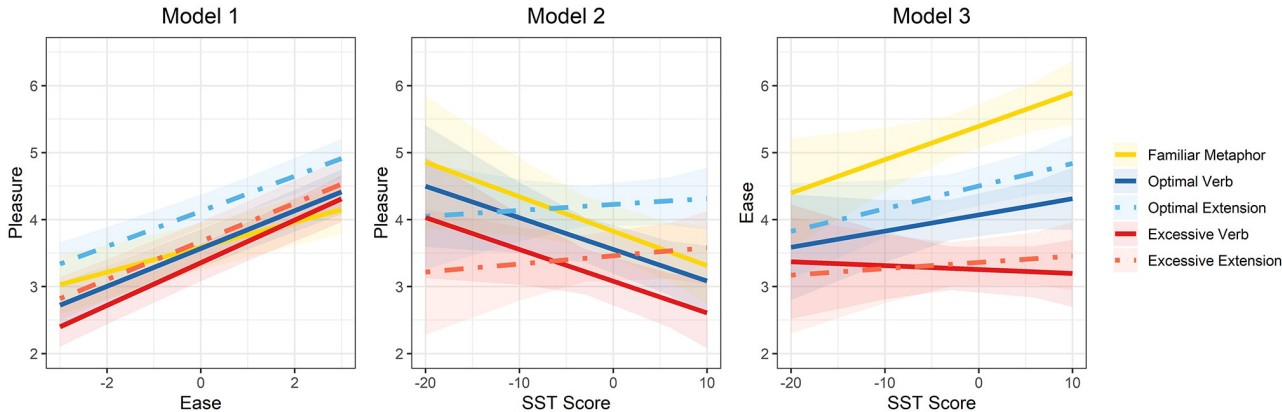

**Fig 3. Experiment 1 model predictions for each variation category (indicated by line style and coloring) with bands showing 95% confidence intervals.**

informative in that it suggests that, unlike single-word changes, innovations that increase context or richness can increase pleasure without decreasing difficulty.

These results may point to the important and widely recognized limitation of using single-sentence stimuli as indicative of the experience of reading complete poems [27]. Although many poems contain individual, strikingly affective phrases, like Gorman's cited above [1], the reading of these phrases is usually shaped by a wealth of both textual and situational context, which will inform readers of how to comprehend a given sentence. Even the reading of extremely short poems—like those in 'haiku' form which are hardly longer than the sentences we tested—will be influenced by situational and/or ecological factors: the presence of a title, preconceptions about the haiku form, even the foreknowledge that the given text constitutes a purposely-written poem. Such contextual factors tend to make comprehension easier and increase pleasure. For instance, studies have shown [e.g., 42] that readers find anomalous metaphorical sentences to be more meaningful if they believe that the sentences are composed by a poet rather than by a computer program, and they will try longer to find them meaningful if a meaning is not readily apparent. This belief would not likely appreciably change how easy a sentence is to understand, but increased 'meaningfulness' might coincide with increased pleasure. In our study, it is possible, that, although the additional words in the optimal extension variations did not make comprehension *easier* than the optimal verb-based variations, they did encourage readers to read in such a way as to find their comprehension pleasurable.

The timing of the variation is also an important difference between the verb and extension variants. In the case of verb variations, only a single word was changed (the verb); this word occurred early in the sentence while also being the word that indicated the metaphorical nature of the sentence. As such, in an example like 'I dash for office', everything depends on the verb 'dash' to indicate simultaneously both the familiar metaphor base (e.g., 'I run for office' = 'I apply to hold elected office') and the variation of it 'run with great haste'. Compare this to the extension, where readers comprehend first the familiar metaphor (e.g., 'I run for office...') and *only after doing so* are given the variation ('...but get tripped up along the way') that encourages the nondefault interpretation. The figurative meaning of the familiar metaphor is likely to be already active when the reader reaches the extension, which can more easily create the pleasurable tension between default and nondefault interpretations. This is particularly intriguing given the general preference toward economy in poetry composition, which would favour the more economical 'I dash for office' over the extension. Further testing and

refinement of the stimulus set regarding the precise length, wording, and placement of extensions compared to equivalent verb variations might help to narrow down the causes of these results (e.g., inclusion of internal, adverbial extensions like 'I run *flat out* for office'), in addition to further manipulation of textual and situational context.

Another area for further refinement of the stimulus set would be to assess not just semantic distances between verbs across conditions and between verbs and nouns within conditions, but the semantic neighborhood densities (SND) of each word (verb or noun) individually and compare them, akin to what was done by Al-Azary and Buchanan [33]. While they found that SND did influence ease of metaphor comprehension, they only compared the SND of nouns in nominal metaphors. In translating their research to our stimuli, consideration would need to be given to whether the individual SND of verbs can be compared directly with those of nouns, or whether the relative SND of verb metaphors like 'I run for office' would be more effectively compared in nominal form (e.g., 'Elections are races'). It should also be noted that Al-Azary and Buchanan indicate the influence of 'concreteness' (vs. 'abstractness') of their nouns [33]. While all the sentences in our stimulus set involves relatively concrete actions (e.g., 'running') interacting with relatively abstract nouns (e.g., 'elected office'), some of those nouns can be considered either 'concrete' or 'abstract' depending on the verb priming and context (e.g., 'office' can be a physical space or an abstract elected position), whereas others are only abstract (e.g., 'meaning'). Further norming of our stimulus set would help to shed additional light on the potential effect of SND and abstractness of target nouns on our findings.

Individual differences, in the form of SST scores, add further complications. Not surprisingly, better recognition of semantic similarities was associated with finding metaphoric sentences easier to understand. But it was also associated with finding them less pleasurable. This (somewhat counterintuitively) suggests that individuals who have more difficulty with metaphor comprehension also find it more pleasurable. Both ease and pleasure ratings of metaphor extensions (both 'optimal' and 'excessive') were less strongly associated with SST performance, suggesting that the kind of verbal reasoning measured by SST is particularly important for shorter metaphors that are particularly dependent on figurative interpretations of single words (verbs, in this case). It is premature, at this point, to make strong inferences based on these data. What is clear, however, is that individual differences in semantic cognition and verbal reasoning (such as those measured by the SST) need to be considered because they strongly affect both ease and pleasure of metaphor comprehension.

In this study, we used two different ways of creating variants of familiar metaphors: changing the critical verb and extending the familiar metaphor with an additional phrase. The verb and extension variants elicited strikingly different responses—the extensions were rated as being more pleasurable (without necessarily being easier to comprehend) and were less sensitive to semantic ability (SST performance). It is possible that (at least some of) the verb variations did not provoke the simultaneous default and nondefault interpretations that should produce pleasure, even in the supposedly optimal condition, and thus that our results only depict the downward trend on the far side of the U-shape. Further refinement of the stimulus set and additional testing might help to clarify this.

A final limitation is that measuring pleasure is an inherently difficult task and likely to be strongly influenced by how the instructions are phrased and how participants interpret them (as discussed above). We tried to be broad in our description of 'pleasure' so as to avoid privileging sentences that described pleasant things over those that described negative things in an effort to shift focus toward the more formal qualities of the sentences themselves. Our instructions undoubtedly privileged a 'poetic' kind of pleasure by suggesting that 'it may help to think of how much you would like to read this sentence in a poem'. Nevertheless, the relatively high pleasure ratings for familiar metaphors—many of which are fairly mundane,

possibly clichéd phrases—suggests that readers were not overly attendant to some expected 'poeticity' of the sentence, which might have predisposed them toward ranking the more obviously 'poetic' verb variations higher.

In future studies, a more fine-grained definition of what we, in this study and following Giora and her team, termed 'pleasure' will help to refine these results. Schindler and her team [37], for instance, provide a broad survey of methods for measuring aesthetic emotions, as well as an Aesthetic Emotions Scale (AESTHEMOS), that may provide further ways of clarifying our definition of pleasure. Kuiken and Douglas, on the other hand, have developed an 'Absorption-like States Questionnaire' (ASQ) [43] intended to help describe readerly activities and aesthetic experiences provoked by literary texts, in particular what they call 'expressive enactment' and 'integrative comprehension'. The former in particular is noted to be relevant for the comprehension of literary metaphors and the production of 'inexpressible' felt states, like what might be characterized as 'resonance', 'meaningfulness' or 'sublime feeling' [44]. Such a questionnaire might allow the maintenance of the 'breadth' of emotions we were seeking to assess while still measuring a degree of 'affectiveness'. Additionally, aligning reported aesthetic experiences with neural activity (e.g., increased sensorimotor simulation [45] or bihemispheric activity [46]) might represent another step in further understanding the curious relationship between optimally difficult metaphors and the feelings they provoke.

## 5. Conclusion

Our results offer only partial support to the hypothesis that, as comprehension difficulty is increased by varying familiar metaphor stimuli (either by changing the verb or extending the metaphor), pleasure will peak at an 'optimal' mid-point level of difficulty. While metaphor extensions appeared to fit this hypothesis, with optimal variation conditions producing more pleasure than the easier familiar or more difficult excessive variation conditions, variations of only the verb did not produce the same effect. Individual differences in the form of SST scores further complicated the picture, indicating that, while increased aptitudes for recognising semantic similarities correlated with reduced difficulty of comprehension across conditions (although more acutely in verb-variation conditions), surprisingly they tended to correlate with reduce pleasure as well. Additional testing, however, will be necessary to strengthen any conclusions regarding the effect of individual differences. Meanwhile, these results also suggest the potential importance of context and variation timing for the pleasure resulting from reading unfamiliar metaphors and indicate several avenues for further research; the stimulus set developed here may provide an important resource for doing so.

## Supporting information

**S1 Fig. In all panels, the points correspond to the behavioural data and the lines correspond to the model fits described in the main text.** The key observation is that none of the panels suggest a U-shape in the behavioural data and the linear models appear to fit the data reasonably well. Left column shows results for familiar metaphors, middle column shows results for optimal verb and optimal extension conditions, right column shows results for excessive verb and excessive extension condition. Top row: relationship between ease of comprehension and pleasure (Model 1). Middle row: relationship between SST Score (semantic knowledge) and pleasure (Model 2). Bottom row: relationship between SST Score (semantic knowledge) and ease of comprehension (Model 3).
(TIF)

## Author Contributions

**Conceptualization:** Patrick J. Errington, Daniel Mirman.

**Data curation:** Melissa Thye, Daniel Mirman.

**Formal analysis:** Melissa Thye.

**Funding acquisition:** Patrick J. Errington.

**Investigation:** Patrick J. Errington, Melissa Thye, Daniel Mirman.

**Methodology:** Patrick J. Errington, Melissa Thye, Daniel Mirman.

**Project administration:** Patrick J. Errington, Daniel Mirman.

**Software:** Melissa Thye.

**Supervision:** Patrick J. Errington, Daniel Mirman.

**Visualization:** Melissa Thye.

**Writing – original draft:** Patrick J. Errington.

**Writing – review & editing:** Patrick J. Errington, Melissa Thye, Daniel Mirman.

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
