## [Decision Letter · Decision Letter 0]

19 Nov 2021

PONE-D-21-32417Difficulty and pleasure in the comprehension of verb-based metaphor sentences: a behavioral studyPLOS ONE

Dear Dr. Errington,

Thank you for submitting your manuscript to PLOS ONE. After careful consideration, we feel that it has merit but does not fully meet PLOS ONE’s publication criteria as it currently stands. As you'll see, the reviewers are quite positive about this study.  And I agree. Your paper is very well-written, the methodology is sound, and the results are potentially interesting. At the same time, the reviewers raised several issues that need to be addressed in a revision. Therefore, we invite you to submit a revised version of the manuscript that addresses the points raised during the review process. Below I summarize some of the main concerns which would need to be addressed.  In addition, each of the reviewers' specific comments needs be addressed (or an explanation provided for not doing so).

First, the logic underlying the use of your individual difference variable (SST) needs to be clarified (see Reviewer 1), as well as the appropriateness of your analyses for examining its role (see Reviewer 2).

Second, as noted by Reviewer 1, there is a substantial literature on the relationship between aesthetic preferences and processing fluency/difficulty that is not referenced in your paper. You should include some of this literature in your revision; doing so may help you clarify the meaning of your results.

Third, some of the analyses you report may be less than optimal for the issues you are investigating (see especially Reviewer 2). As noted by Reviewer 2, examining the ease-pleasure relationship separately for the different categories may result in the hypothesized curvilinear relationship being obscured.  Why not examine the ease-pleasure relationship across the entire data set? As well, there appears to be some redundancy in some of the models you test because you simultaneously include comprehension ease as well as metaphor category (which is based, in part, on comprehension ease).

Fourth, you should describe how your sample size was determined and address the issue of power to detect your predicted effects.

Fifth, participants in your experiment provided four ratings.  However, the analyses you report include only two (pleasure and ease).  I assume you’ve undertaken analyses with the other two rating scales (imageability and emotion) and some mention should be made of them (at the very least in a footnote).

We look forward to receiving your revised manuscript.

Kind regards,

Thomas Holtgraves, Ph.D.

Academic Editor

PLOS ONE

Journal Requirements:

4. Please remove your figures from within your manuscript file, leaving only the individual TIFF/EPS image files, uploaded separately.  These will be automatically included in the reviewers’ PDF.

Reviewers' comments:

Reviewer's Responses to Questions

**Comments to the Author**

1. Is the manuscript technically sound, and do the data support the conclusions?

Reviewer #1: Partly

Reviewer #2: Partly

2. Has the statistical analysis been performed appropriately and rigorously? 

Reviewer #1: Yes

Reviewer #2: No

3. Have the authors made all data underlying the findings in their manuscript fully available?

Reviewer #1: Yes

Reviewer #2: Yes

4. Is the manuscript presented in an intelligible fashion and written in standard English?

Reviewer #1: Yes

Reviewer #2: Yes

5. Review Comments to the Author

Reviewer #1: 1. It is difficult to determine what kind of “pleasure” the authors are trying to address. The terminology varies: striking, thrilling, affective, pleasurable, etc. There is a substantial psychometric issue lurking here that should be addressed more directly. The authors may find it useful to locate “effective, satisfying, or powerful” (the wording eventually chosen for their ratings; pp. 18-19) within the most comprehensive available survey of aesthetic “emotions” (Schindler et al. 2017). Although the author’s formalist intent is explicit (“how much you liked the way the message was expressed”; l. 434), the kind of “affectiveness” that is at stake also remains obscure.

Schindler, I., Hosoya, G., Menninghaus, W., Beermann, U., Wagner, V., Eid, M., & Scherer, K. R. (2017). Measuring aesthetic emotions: A review of the literature and a new assessment tool. PLOS ONE, 12(6), e0178899. https://doi.org/10.1371/journal.pone.017889

2. The authors indicate that their research is “informed” by Giora’s “optimal innovation hypothesis” (see especially, Giora et al. 2017).

a. One issue in Giora’s work is how to differentiate default literal, default metaphoric, and non-default metaphoric sentences so that metaphoric default salience can be assessed independently of metaphoric non-default salience. It is somewhat disconcerting, then, to learn that the authors did not include the default literal sentences in Study 1 because “we were only interested in metaphor comprehension” (l. 368). Giora is primarily interested in metaphor comprehension, too; so, the reason for ignoring differences between default literal and default metaphoric sentences should be more clearly articulated—and in terms she would understand.

b. A related but separate issue is how to differentiate metaphoric sentences so that, independently of defaultness, metaphoric vehicles can be contrasted according to the extent to which they are “domain-specific” (l. 167). The authors hypothesize that excessive domain specificity may make a metaphor too difficult to be pleasurable. However, emphasizing domain-specificity may oversimplify the problem. For example, Katz and Al-Azary (2017) differentiate (a) the distance between the domain of the topic and the domain of the vehicle; (b) the semantic density of the topic and of the vehicle; and (c) the specificity of the topic or vehicle within their respective domains (“domain-specificity”?). The authors should be encouraged to compare the Katz and Al-Zahry framework with their own. For example, perhaps they can clarify whether their use of word2vec (l. 267) converges with the Katz and Al-Zahry version of computational semantics.

Katz, A. N., & Al-Azary, H. (2017). Principles that promote bidirectionality in verbal metaphor. Poetics Today, 38(1), 35–59. https://doi.org/10.1215/03335372-3716215

c. Contextualizing their hypothesis within the Katz and Alzary (2017) framework may also enable the authors to clarify their rationale for including the Semantic Similarities Test (SST) as an individual differences measure of “the ability to identify conceptual mappings between words” (l. 181). The SST assesses “crystalized verbal intelligence” by evaluating a person’s ability to find similarities between two concepts. Was this measure expected to reflect (a) the capacity to identify similarities between vehicle and topic concepts even when they are “domain-specific”; (b) the capacity to identify similarities between vehicle and topic concepts even when they are from distant domains; or (c) the capacity to identify similarities between vehicle and topic concepts because they are (especially for some individuals) semantically dense? As it stands, the SST is not conceptually well coordinated with the author’s research paradigm.

d. It is a bit unsettling that there is no reference to other literature indicating that aesthetic preferences are determined by processing fluency/difficulty (cf. Reber et al., 2004), including the tradition of proposed curvilinear relations between object complexity and interest/pleasure (e.g., Berlyne, 1971). Research in the latter tradition substantiates how difficult it is to assess curvilinear relations beccause the direction of the relationship differs at different levels of the variables. More to the point, it is not clear where to locate “optimal” and “excessive” metaphors on the hypothesized curvilinear relation with pleasure. That difficulty should at least be mentioned.

3. Perhaps the most innovative aspect of the authors’ design is their attempt to examine novel extensions of familiar metaphors. In this reviewer’s judgment, their procedures for doing so are promising. The procedures used to develop these extended metaphors led to at least two interesting results. First, optimal extensions and excessive extensions were rated as the “most figurative” (l. 358). Second, metaphor extensions were rated as more pleasurable than verb variants, indicating that “innovations that increase … richness can increase pleasure without decreasing difficulty. These results are worth building on in future research efforts.

a. To explain these results, the authors emphasize “textual and situational context.” However, it may be more promising to examine specifically extended metaphors. The authors offer the following hypothesis: “The figurative meaning of the familiar metaphor is likely to be already active when the reader reaches the extension, which can more easily create the pleasurable tension between default and non-default interpretations.” The possibility of examining the interplay (and tension) between a familiar metaphor and a subsequent extension is within these authors’ methodological reach.

b. As their project unfolds, the authors may want to take advantage of Sullivan’s (2019) recent examination of mixed metaphors, some of which are subject to the domain specificity problem that makes them anathema in scholarly circles—but perhaps of particular interest to the authors.

Sullivan, K. (2018). Mixed metaphors: Their use and abuse. Bloomsbury Publishing.

4. A not-so-important note (l. 290): a 7-point rating scale is not a “Likert scale.” This common misuse of that phrase should not be repeated here.

Reviewer #2: In this paper, the authors examine the “optimal innovation hypothesis”, which posits that language is most enjoyable when it evokes a non-default response, but also brings to mind the default response so that the two responses can be weighted against one another. The authors examined this hypothesis by carefully constructing a stimulus set of metaphors and obtaining subjective ratings of pleasurability for these metaphors.

First, the paper was well-written. It discussed an interesting theory within figurative language research. Also, I really enjoyed the writing style. I thought the examples used were interesting and made the paper exciting, and the authors also explained the theories and hypotheses clearly.

Second, I think the stimuli for this study are very nice, and I appreciate that the authors made them available on OSF. The authors carefully considered a variety of factors, including semantic distance measures and frequency. I also totally agree with the authors that there is an overemphasis on examining nominal metaphors, and I appreciate that they developed a set of verb metaphors and metaphor extensions and made these publicly available. I was very impressed by this aspect of the study.

That being said, I have major concern with the analyses the authors conducted to examine their hypotheses (Section 3.2: Results and Discussion). I question the logic of these analyses for a couple reasons. First of all, the authors mention that there was no U-shaped curve observed in the data. However, to examine this, they split the familiar, optimal, and excessive metaphors into separate categories and examined these curves separately (Supplementary Figure). I think this defeats the purpose as this restricts the range of the data in each case. The whole point of the optimal and excessive categories was that they were supposed to increase difficulty, so it doesn’t make sense to me why difficulty would be examined within each of these categories separately. Wouldn’t it make more sense to either 1) look at the relationship between ease and pleasure collapsed across all the metaphors, or 2) compare the means between the categories directly (metaphors in the optimal category should have significantly higher pleasurability ratings than the familiar and excessive categories)? This criticism carries through to the first lme model (Model 1). Both the categories and the continuous ratings of ease were included in the model, but aren’t these essentially the same variable? Therefore, (if I understand lme correctly) if there is a significant effect of a category, this effect is independent of ease since ease is a variable in the model. However, the categories were specifically constructed to vary in ease, so the critical variable in the category is being controlled out of the category. Also, I didn’t quite get the logic of Model 2. It considered individual differences, but the relative differences of ease between the metaphors should still be preserved regardless of individual differences in figurative comprehension, so I’m also not sure how that tests the hypothesis.

In the discussion, the authors state “the optimal metaphor extensions were rated as intermediate in terms of ease of comprehension, but highest in terms of pleasure”. And also, “familiar metaphors were rated both easier and more pleasurable than ‘optimal’ verb variants, which were rated both easier and more pleasurable than ‘excessive’ verb variants”. But these comparisons were not tested statistically, and weren’t really discussed much in the Results. I think the lme models obscure these findings, which are probably the most interesting part of the data.

I think this is a fixable problem though. I would recommend just conducting some standard statistics on the data, namely, an ANOVA between the categories (not considering ease ratings) and a correlation between ease and pleasurability (independent of the categories). It might be beneficial to run some separate ANOVAs to compare both optimal vs. excessive, but also verb vs. extension, since there seemed to be a trend in extensions being more pleasurable. I think this would more directly test your hypotheses and would make the Results section better fit with the Discussion section.

As one additional small point, I didn’t understand the NGram frequency graph on page 14. I think the axes need to be labelled or else the graph needs to be better explained in the caption.

Overall, I think this paper has a lot of potential. The stimuli were very carefully constructed, and the data seems quite interesting. However, in my opinion, I think the analyses were somewhat inappropriate for the reasons stated above.

6. PLOS authors have the option to publish the peer review history of their article (what does this mean?). If published, this will include your full peer review and any attached files.

Reviewer #1: No

Reviewer #2: No

---

## [Author Response · Author response to Decision Letter 0]

3 Jan 2022

Academic Editor Comments

1) the logic underlying the use of your individual difference variable (SST) needs to be clarified (see Reviewer 1), as well as the appropriateness of your analyses for examining its role (see Reviewer 2). 

We thank the editor and both reviewers for the suggestion to further clarify the Semantic Similarities Test and its analysis. We have explained the importance of the SST as a means of ascertaining the effects of an individual’s capacity for metaphorical thinking, which we hypothesised would impact the ease with which a given individual would be able to resolve all the variations of metaphors with which they were presented. This was explained with regard to Reviewer 1’s comments (ll. 224–233). This hypothesis was borne out in our study, which found those with higher SST scores to have greater ease in comprehending all categories of metaphor; surprisingly, however, those with higher SST scores did not show greater pleasure in comprehending the ‘optimal extension’ condition, nor did they show an increase in pleasure in comprehending the ‘excessive extension’, which we would have expected if their increased metaphor comprehension capacities had only moved the apex of the inverted U-shaped relationship toward the more challenging phrases. This effect is addressed in our discussion (ll. 693–707)

2) as noted by Reviewer 1, there is a substantial literature on the relationship between aesthetic preferences and processing fluency/difficulty that is not referenced in your paper. You should include some of this literature in your revision; doing so may help you clarify the meaning of your results.

We greatly appreciate reviewer 1’s suggested literature regarding aesthetic preference and processing fluency, and we found it quite helpful in framing our study (see additions ll. 113¬–132).

3) some of the analyses you report may be less than optimal for the issues you are investigating (see especially Reviewer 2). As noted by Reviewer 2, examining the ease-pleasure relationship separately for the different categories may result in the hypothesized curvilinear relationship being obscured. Why not examine the ease-pleasure relationship across the entire data set? As well, there appears to be some redundancy in some of the models you test because you simultaneously include comprehension ease as well as metaphor category (which is based, in part, on comprehension ease).

We appreciate the recommendation for a simpler analysis approach, which we have added to the manuscript. A more detailed description is included below in our response to the reviewer’s comment. We have also retained our original analyses because we believe they provide useful additional insight into the results.

4) you should describe how your sample size was determined and address the issue of power to detect your predicted effects.

We have clarified (ll. 458–463) that due to the novelty of this stimulus set and our experiment design, there was not a strong basis for predicting an effect size, which is necessary for a proper power analysis. Instead, we determined the sample size based on sample sizes from prior metaphor comprehension studies and practical limitations. We hope that, by making available our full stimulus set and analysis code, this study will provide the basis for future power calculations.

5) participants in your experiment provided four ratings. However, the analyses you report include only two (pleasure and ease). I assume you’ve undertaken analyses with the other two rating scales (imageability and emotion) and some mention should be made of them (at the very least in a footnote).

Those other two ratings (imageability and emotion) were included for later exploratory analyses, which have not been conducted yet. Because we did not have clear hypotheses about them and they are not directly related to the hypotheses evaluated here, we feel that they are outside the scope of this manuscript and have not added them. We only mention that those ratings were collected in the Methods for the purpose of transparency.

 

Reviewer 1

1. It is difficult to determine what kind of “pleasure” the authors are trying to address. The terminology varies: striking, thrilling, affective, pleasurable, etc. There is a substantial psychometric issue lurking here that should be addressed more directly. The authors may find it useful to locate “effective, satisfying, or powerful” (the wording eventually chosen for their ratings; pp. 18-19) within the most comprehensive available survey of aesthetic “emotions” (Schindler et al. 2017). Although the author’s formalist intent is explicit (“how much you liked the way the message was expressed”; l. 434), the kind of “affectiveness” that is at stake also remains obscure.

Schindler, I., Hosoya, G., Menninghaus, W., Beermann, U., Wagner, V., Eid, M., & Scherer, K. R. (2017). Measuring aesthetic emotions: A review of the literature and a new assessment tool. PLOS ONE, 12(6), e0178899. https://doi.org/10.1371/journal.pone.017889

This point is well noted, and we thank the reviewer for these recommendations. We have sought to clarify (ll. 509–517) that the increase in aesthetic response (which we have, perhaps overly simply, termed ‘pleasure’) is more precisely an increase of affect in a broad sense rather than one or another particular ‘aesthetic emotion’ like those surveyed by Schindler et al. In our case, we sought to test whether whatever ‘content’ feeling was elicited by a familiar sentence could be increased or intensified by the variations.

2. The authors indicate that their research is “informed” by Giora’s “optimal innovation hypothesis” (see especially, Giora et al. 2017).

a) One issue in Giora’s work is how to differentiate default literal, default metaphoric, and non-default metaphoric sentences so that metaphoric default salience can be assessed independently of metaphoric non-default salience. It is somewhat disconcerting, then, to learn that the authors did not include the default literal sentences in Study 1 because “we were only interested in metaphor comprehension” (l. 368). Giora is primarily interested in metaphor comprehension, too; so, the reason for ignoring differences between default literal and default metaphoric sentences should be more clearly articulated—and in terms she would understand.

We again thank the reviewer for their probing questions, and we have sought to explain (ll. 429–439) that, where for Giora et al., the ‘default’ meaning is primarily non-metaphoric with a variation that introduces a metaphoric/figurative component (one of her examples is ‘A piece of paper’ which becomes ‘A peace of paper’), in our case, the ‘default’ meaning from the phrase was purely metaphoric. In ‘I run for office’, there is (according to studies and surveyed in Holyoak and Stamenkovic 2017) little recourse to any literal running, whereas we hypothesise that in ‘I dash for office’ there may be a dual activation of both the metaphorical meaning (‘I seek an elected office’) and some aspect of a literal meaning (‘I run quickly to achieve something physical). In this case the familiar metaphors provide an optimal control condition because they can be matched to the variations in terms of general meaning, figurativeness, and basic psycholinguistic properties (length, word frequency, etc.). 

b) A related but separate issue is how to differentiate metaphoric sentences so that, independently of defaultness, metaphoric vehicles can be contrasted according to the extent to which they are “domain-specific” (l. 167). The authors hypothesize that excessive domain specificity may make a metaphor too difficult to be pleasurable. However, emphasizing domain-specificity may oversimplify the problem. For example, Katz and Al-Azary (2017) differentiate (a) the distance between the domain of the topic and the domain of the vehicle; (b) the semantic density of the topic and of the vehicle; and (c) the specificity of the topic or vehicle within their respective domains (“domain-specificity”?). The authors should be encouraged to compare the Katz and Al-Zahry framework with their own. For example, perhaps they can clarify whether their use of word2vec (l. 267) converges with the Katz and Al-Zahry version of computational semantics.

Katz, A. N., & Al-Azary, H. (2017). Principles that promote bidirectionality in verbal metaphor. Poetics Today, 38(1), 35–59. https://doi.org/10.1215/03335372-3716215

The reviewer’s suggestion to relate our work to that of Katz and Al-Azary is much appreciated and we have expanded our discussion regarding ‘domain-specificity’ to incorporate some of the ‘semantic density’ language offered by Katz and Al-Azary (ll. 199–208). We have specified that the word2vec norming does indeed converge with their computational methods for determining that density. This is also noted in the ‘Objective Measures’ subsection of the ‘Stimulus Development’ section (ll. 321–322).

c) Contextualizing their hypothesis within the Katz and Alzary (2017) framework may also enable the authors to clarify their rationale for including the Semantic Similarities Test (SST) as an individual differences measure of “the ability to identify conceptual mappings between words” (l. 181). The SST assesses “crystalized verbal intelligence” by evaluating a person’s ability to find similarities between two concepts. Was this measure expected to reflect (a) the capacity to identify similarities between vehicle and topic concepts even when they are “domain-specific”; (b) the capacity to identify similarities between vehicle and topic concepts even when they are from distant domains; or (c) the capacity to identify similarities between vehicle and topic concepts because they are (especially for some individuals) semantically dense? As it stands, the SST is not conceptually well coordinated with the author’s research paradigm.

Further discussion of the Semantic Similarities Test and its hypothesised and tested role in affecting metaphor phrase processing ease/difficulty has been incorporated (ll. 224–233), building upon discussion of Katz and Al-Alzary’s frameworks (see point above).

d) It is a bit unsettling that there is no reference to other literature indicating that aesthetic preferences are determined by processing fluency/difficulty (cf. Reber et al., 2004), including the tradition of proposed curvilinear relations between object complexity and interest/pleasure (e.g., Berlyne, 1971). Research in the latter tradition substantiates how difficult it is to assess curvilinear relations because the direction of the relationship differs at different levels of the variables. More to the point, it is not clear where to locate “optimal” and “excessive” metaphors on the hypothesized curvilinear relation with pleasure. That difficulty should at least be mentioned.

The reviewer’s suggestion to relate our work more explicitly to the history of proposed curvilinear relations between complexity and interest/pleasure is greatly appreciated. An expanded discussion of this history (ll. 113–132) has been included, leading to further discussions of the SST (ll. 216–233), in which we reference how our hypotheses contradict the predominantly linear hypotheses regarding processing fluency (e.g., Reber et al., 2004), wherein the difficulties in locating our stimulus categories along the hypothesized inverted U-shape relationship are indicated in response to a previous suggestion.

3. Perhaps the most innovative aspect of the authors’ design is their attempt to examine novel extensions of familiar metaphors. In this reviewer’s judgment, their procedures for doing so are promising. The procedures used to develop these extended metaphors led to at least two interesting results. First, optimal extensions and excessive extensions were rated as the “most figurative” (l. 358). Second, metaphor extensions were rated as more pleasurable than verb variants, indicating that “innovations that increase … richness can increase pleasure without decreasing difficulty. These results are worth building on in future research efforts.

We thank the reviewer for this positive appraisal of our procedures and results, and their encouragement for future research efforts.

a) To explain these results, the authors emphasize “textual and situational context.” However, it may be more promising to examine specifically extended metaphors. The authors offer the following hypothesis: “The figurative meaning of the familiar metaphor is likely to be already active when the reader reaches the extension, which can more easily create the pleasurable tension between default and non-default interpretations.” The possibility of examining the interplay (and tension) between a familiar metaphor and a subsequent extension is within these authors’ methodological reach.

b) As their project unfolds, the authors may want to take advantage of Sullivan’s (2019) recent examination of mixed metaphors, some of which are subject to the domain specificity problem that makes them anathema in scholarly circles—but perhaps of particular interest to the authors.

Sullivan, K. (2018). Mixed metaphors: Their use and abuse. Bloomsbury Publishing.

We wish again to express our thanks the reviewer for these two thoughtful suggestions for future areas of investigation.

4. A not-so-important note (l. 290): a 7-point rating scale is not a “Likert scale.” This common misuse of that phrase should not be repeated here.

We have removed the term ‘Likert’ as requested.

 

Reviewer 2

In this paper, the authors examine the “optimal innovation hypothesis”, which posits that language is most enjoyable when it evokes a non-default response, but also brings to mind the default response so that the two responses can be weighted against one another. The authors examined this hypothesis by carefully constructing a stimulus set of metaphors and obtaining subjective ratings of pleasurability for these metaphors.

First, the paper was well-written. It discussed an interesting theory within figurative language research. Also, I really enjoyed the writing style. I thought the examples used were interesting and made the paper exciting, and the authors also explained the theories and hypotheses clearly.

Second, I think the stimuli for this study are very nice, and I appreciate that the authors made them available on OSF. The authors carefully considered a variety of factors, including semantic distance measures and frequency. I also totally agree with the authors that there is an overemphasis on examining nominal metaphors, and I appreciate that they developed a set of verb metaphors and metaphor extensions and made these publicly available. I was very impressed by this aspect of the study.

That being said, I have major concern with the analyses the authors conducted to examine their hypotheses (Section 3.2: Results and Discussion). I question the logic of these analyses for a couple reasons. 

1) First of all, the authors mention that there was no U-shaped curve observed in the data. However, to examine this, they split the familiar, optimal, and excessive metaphors into separate categories and examined these curves separately (Supplementary Figure). I think this defeats the purpose as this restricts the range of the data in each case. The whole point of the optimal and excessive categories was that they were supposed to increase difficulty, so it doesn’t make sense to me why difficulty would be examined within each of these categories separately. Wouldn’t it make more sense to either 1) look at the relationship between ease and pleasure collapsed across all the metaphors, or 2) compare the means between the categories directly (metaphors in the optimal category should have significantly higher pleasurability ratings than the familiar and excessive categories)? This criticism carries through to the first lme model (Model 1). Both the categories and the continuous ratings of ease were included in the model, but aren’t these essentially the same variable? Therefore, (if I understand lme correctly) if there is a significant effect of a category, this effect is independent of ease since ease is a variable in the model. However, the categories were specifically constructed to vary in ease, so the critical variable in the category is being controlled out of the category. Also, I didn’t quite get the logic of Model 2. It considered individual differences, but the relative differences of ease between the metaphors should still be preserved regardless of individual differences in figurative comprehension, so I’m also not sure how that tests the hypothesis.

In the discussion, the authors state “the optimal metaphor extensions were rated as intermediate in terms of ease of comprehension, but highest in terms of pleasure”. And also, “familiar metaphors were rated both easier and more pleasurable than ‘optimal’ verb variants, which were rated both easier and more pleasurable than ‘excessive’ verb variants”. But these comparisons were not tested statistically, and weren’t really discussed much in the Results. I think the lme models obscure these findings, which are probably the most interesting part of the data.

I think this is a fixable problem though. I would recommend just conducting some standard statistics on the data, namely, an ANOVA between the categories (not considering ease ratings) and a correlation between ease and pleasurability (independent of the categories). It might be beneficial to run some separate ANOVAs to compare both optimal vs. excessive, but also verb vs. extension, since there seemed to be a trend in extensions being more pleasurable. I think this would more directly test your hypotheses and would make the Results section better fit with the Discussion section.

We thank the reviewer for suggesting a simpler and more straightforward analysis strategy. Looking at the ease-pleasure relationship collapsed across qualitatively different types of metaphors would create the possibility of Simpson’s Paradox, so we have followed the second suggestion and now include simpler analyses of (i) Ease differences between conditions (to validate the experimental manipulation) and (ii) Pleasure differences between conditions (to test the critical hypotheses). We kept the analyses within the LME framework, though these simpler analyses are analogous to a repeated-measures ANOVA (LME allows more flexible specification of crossed random effects of participants and items). These analyses show monotonic differences in ease (Familiar > Optimal > Excessive) for both verb and extension manipulations. For pleasure ratings, there is a U-shape for extensions, but not for verbs.

Although the metaphor types differ in average ease of comprehension, there is substantial overlap and the range of ease values is very similar across the different types. So in addition to the simpler model suggested by the reviewer, we tested the more complex model that includes both variables (ease and metaphor type) – it allows testing whether the ease-pleasure relationship holds within categories and differs between categories.

As one additional small point, I didn’t understand the NGram frequency graph on page 14. I think the axes need to be labelled or else the graph needs to be better explained in the caption.

Thank for bringing this to our attention. We have elaborated the explanation in the Figure 1 caption. 

Overall, I think this paper has a lot of potential. The stimuli were very carefully constructed, and the data seems quite interesting. However, in my opinion, I think the analyses were somewhat inappropriate for the reasons stated above.

---

## [Decision Letter · Decision Letter 1]

21 Jan 2022

PONE-D-21-32417R1Difficulty and pleasure in the comprehension of verb-based metaphor sentences: a behavioral studyPLOS ONE

Dear Dr. Errington,

Thank you for submitting your revised manuscript to PLOS ONE.  I sent your revision to the two original reviewers and both believed your revisions to be quite responsive to their comments.  And I agree. However, they also raise a couple of final issues that I'd like you to address before your manuscript can be accepted for publication.  Both reviewers ask for additional clarification of Katz and Al-Azary's (2017) Semantic Neighborhood Density (SND) measure, and Reviewer 1 suggests providing some guidance for future research regarding the psychometric issues involved in assessing your construct.

We look forward to receiving your revised manuscript.

Kind regards,

Thomas Holtgraves, Ph.D.

Academic Editor

PLOS ONE

Journal Requirements:

Reviewers' comments:

Reviewer's Responses to Questions

**Comments to the Author**

1. If the authors have adequately addressed your comments raised in a previous round of review and you feel that this manuscript is now acceptable for publication, you may indicate that here to bypass the “Comments to the Author” section, enter your conflict of interest statement in the “Confidential to Editor” section, and submit your "Accept" recommendation.

Reviewer #1: All comments have been addressed

Reviewer #2: (No Response)

2. Is the manuscript technically sound, and do the data support the conclusions?

Reviewer #1: Yes

Reviewer #2: Yes

3. Has the statistical analysis been performed appropriately and rigorously? 

Reviewer #1: Yes

Reviewer #2: Yes

4. Have the authors made all data underlying the findings in their manuscript fully available?

Reviewer #1: Yes

Reviewer #2: Yes

5. Is the manuscript presented in an intelligible fashion and written in standard English?

Reviewer #1: Yes

Reviewer #2: Yes

6. Review Comments to the Author

Reviewer #1: I indicated that all of the present reviewer's comments have been addressed, but I still have qualms about how effectively two of those comments were addressed. First, although it is not possible to undo the difficulties created by a psychometrically thin and conceptually weak articulation of "affectiveness" (l. 94), "meaningfulness" (l. 639) etc., the authors would do well to say something about the steps that might be taken in future efforts to provide psychometric substance to the construct they are trying to assess. References to Shklovsky and Miall are not enough (e.g., is "strikingness" the same as "affectiveness"? how would we know?). Very briefly, please say how this psychometric issue might be addressed in future studies. Second, it is not clear that the authors have precisely coordinated their use of word2vec with the the DIFFERENTIATION that Katz and Al-Zary (2017) offer between (a) the distance between the vehicle and topic semantic neighborhoods and (b) the density of the separately considered vehicle and topic (l. 301). Is it possible to more precise about how these two aspects of metaphoric semantic structures are related to the procedures they rely on?

Reviewer #2: The authors have addressed the points I brought up in the last review. I think the manuscript looks excellent and will be very interesting to the PLoS ONE readership.

There is one small point of clarification about Katz and Al-Azary's (2017) Semantic Neighborhood Density (SND) measure. It is not exactly synonymous to semantic distance where two terms are compared, but rather, it is a measure of the average distance from a single term to its own neighbors. So, for a metaphor like "ski for office", SND doesn't actually measure the overlap between ski and office. Rather, it measures the semantic density of "ski" on its own and "office" on its own. If the sentence was "ski downhill", the SND for "ski" is still the exact same. It may indirectly get at overlap -- if "ski" and "office" both have high SND, then they may have more overlapping near neighbors, but SND on its own is not a measure of this. Just wanted to clarify this because the way it is written in the manuscript, it sounds like SND is a measure of the overlapping semantic neighbors between two terms.

They explain Semantic Neighborhood Density in a little more detail in this paper:

Al-Azary, H., McAuley, T., Buchanan, L., & Katz, A. N. (2019). Semantic processing of metaphor: A case-study of deep dyslexia. Journal of Neurolinguistics, 51, 297-308.

Thank you very much for the opportunity to review this work!

7. PLOS authors have the option to publish the peer review history of their article (what does this mean?). If published, this will include your full peer review and any attached files.

Reviewer #1: No

Reviewer #2: No

---

## [Author Response · Author response to Decision Letter 1]

25 Jan 2022

Reviewer #1: I indicated that all of the present reviewer's comments have been addressed, but I still have qualms about how effectively two of those comments were addressed. First, although it is not possible to undo the difficulties created by a psychometrically thin and conceptually weak articulation of "affectiveness" (l. 94), "meaningfulness" (l. 639) etc., the authors would do well to say something about the steps that might be taken in future efforts to provide psychometric substance to the construct they are trying to assess. References to Shklovsky and Miall are not enough (e.g., is "strikingness" the same as "affectiveness"? how would we know?). Very briefly, please say how this psychometric issue might be addressed in future studies. 

Review 1’s suggestions to further explain how, in future experiments, might refine our definition of ‘aesthetic pleasure’, which were, following Giora et al., perhaps overbroad, are much appreciated. We have, in response, added a final note to our Discussion section indicating the possibilities of using both Schindler et al.’s survey and Aesthetic Emotions Scale and Kuiken and Douglas’s Absorption-like States Questionnaire to clarify this definition and refine our questions for future experiments. See lines 747–765.

Second, it is not clear that the authors have precisely coordinated their use of word2vec with the the DIFFERENTIATION that Katz and Al-Zary (2017) offer between (a) the distance between the vehicle and topic semantic neighborhoods and (b) the density of the separately considered vehicle and topic (l. 301). Is it possible to more precise about how these two aspects of metaphoric semantic structures are related to the procedures they rely on?

Reviewer #2: The authors have addressed the points I brought up in the last review. I think the manuscript looks excellent and will be very interesting to the PLoS ONE readership.

There is one small point of clarification about Katz and Al-Azary's (2017) Semantic Neighborhood Density (SND) measure. It is not exactly synonymous to semantic distance where two terms are compared, but rather, it is a measure of the average distance from a single term to its own neighbors. So, for a metaphor like "ski for office", SND doesn't actually measure the overlap between ski and office. Rather, it measures the semantic density of "ski" on its own and "office" on its own. If the sentence was "ski downhill", the SND for "ski" is still the exact same. It may indirectly get at overlap -- if "ski" and "office" both have high SND, then they may have more overlapping near neighbors, but SND on its own is not a measure of this. Just wanted to clarify this because the way it is written in the manuscript, it sounds like SND is a measure of the overlapping semantic neighbors between two terms.

They explain Semantic Neighborhood Density in a little more detail in this paper:

Al-Azary, H., McAuley, T., Buchanan, L., & Katz, A. N. (2019). Semantic processing of metaphor: A case-study of deep dyslexia. Journal of Neurolinguistics, 51, 297-308.

Thank you very much for the opportunity to review this work!

We thank both the reviewers for their requests and suggestions for additional clarification on the potential relevance of semantic density (SND) to our study. We have removed our erroneous suggestion that our use of word2vec aligned with semantic density in our ‘Objective Measures’ subsection (l. 320–322 – formerly l. 301), as this was truly designed to measure distance between nouns and their interactive verbs within each condition, and between critical verbs between conditions.

A brief discussion of Al-Azary and Buchanan’s (2017) study of the influence of SND’s on novel metaphor comprehension is included in the introduction section (ll. 245–260). This replaces the less-appropriate discussion of Katz and Al-Azary (2017) (formerly l. 188). This article’s examination of SND and the bidirectionality of metaphors seems, upon reflection, to be less directly relevant than the Al-Azary and Buchanan’s (2017) examination of SND, concreteness, and metaphor comprehensibility. 

A discussion of the possible influence of SND and difficulties of calculating and comparing the SND in our verb-metaphor stimulus set has been added to the Discussion section, lines 688–707.

---

## [Editor Report · Decision Letter 2]

27 Jan 2022

Difficulty and pleasure in the comprehension of verb-based metaphor sentences: a behavioral study

PONE-D-21-32417R2

Dear Dr. Errington,

We’re pleased to inform you that your manuscript has been judged scientifically suitable for publication and will be formally accepted for publication once it meets all outstanding technical requirements.

Kind regards,

Thomas Holtgraves, Ph.D.

Academic Editor

PLOS ONE
---

## [Editor Report · Acceptance letter]

2 Feb 2022

PONE-D-21-32417R2 

Difficulty and pleasure in the comprehension of verb-based metaphor sentences: A behavioral study 

Dear Dr. Errington:

I'm pleased to inform you that your manuscript has been deemed suitable for publication in PLOS ONE. Congratulations! Your manuscript is now with our production department. 

Kind regards, 

on behalf of

Dr. Thomas Holtgraves 

Academic Editor

PLOS ONE